# Structural-Health-Monitoring-Oriented Finite Element Model for a Specially Shaped Steel Arch Bridge and Its Application

**Li Dai** [1], **Mi-Da Cui** [2] and **Xiao-Xiang Cheng** [2,*]

1  Jiangxi Transportation Institute Co., Ltd., Nanchang 330200, China
2  School of Civil Engineering, Southeast University, Nanjing 211189, China
*  Correspondence: cxx_njut@hotmail.com

**Abstract:** To rigorously evaluate the health of a steel bridge subjected to vehicle-induced fatigue, both a detailed numerical model and effective fatigue analysis methods are needed. In this paper, the process for establishing the structural health monitoring (SHM)-oriented finite element (FE) model and assessing the vehicle-induced fatigue damage is presented for a large, specially shaped steel arch bridge. First, the bridge is meticulously modeled using multiple FEs to facilitate the exploration of the local structural behavior. Second, manual tuning and model updating are conducted according to the modal parameters measured at the bridge's location. Since the numerical model comprises a large number of FEs, two surrogate-model-based methods are employed to update the model. Third, the established models are validated by using them to predict the structure's mode shapes and the actual structural behavior for the case in which the whole bridge is subjected to static vehicle loads. Fourth, using the numerical model, a new fatigue analysis method based on the high-circle fatigue damage accumulation theory is employed to further analyze the vehicle-induced fatigue damage to the bridge. The results indicate that manual tuning and model updating are indispensable for SHM-oriented FE models with erroneous configurations, and one surrogate-model-based model updating method is effective. In addition, it is shown that the fatigue analysis method based on the high-circle fatigue damage accumulation theory is applicable to real-world engineering cases.

**Keywords:** arch bridge; finite element model; model updating; response surface method; multi-output support vector regression method; fatigue analysis; high-circle fatigue damage accumulation theory

## 1. Introduction

Steel bridges respond to vehicle loads every day, possibly leading to fatigue and other local structural damages. To ensure their safety, structural health monitoring (SHM) systems are installed on many of them. Nearly every SHM system employs a finite element (FE) model for its condition assessments and structural behavior predictions. According to Fei et al. [1], Duan et al. [2], and Cheng et al. [3], these models should be as meticulous as possible for two main reasons. First, local defects and other local structural behaviors, such as fatigue under different loading conditions, can be explored via an SHM system. Second, model updating can be undertaken without considering additional concerns on section properties, assuming they have been satisfactorily modeled with all the geometric details fully taken into account.

When the structural information is measured at the location, model updating is generally carried out for the established numerical models to ensure the quality of the simulation. The well-established method for updating FE models is the sensitivity-based parametric updating approach that directly uses the numerical models [1,4,5]. However, a performance problem arises with updating these meticulous FE models using the traditional method: when all of the geometric details are embraced, the models grow in complexity, and model updating becomes infeasible directly using the FE models due to the high

computational cost. Fortunately, some simple surrogate models can replace the FE model in this situation. Currently, two surrogate-model-based methods have been proposed in the field of FE model updating, i.e., the response surface (RS) method and the multi-output support vector regression (MSVR) method. Deng and Cai [6], Ren et al. [7], Ren and Chen [8], and Zhou et al. [9] demonstrated the efficiency and effectiveness of the RS method in bridge model updating. According to Teng et al. [10], the MSVR method is also effective for the FE model updating of a large-span spatial steel structure. However, which method is practically superior to the other in updating an FE model based on surrogate models? This significant question of practical importance has not yet been answered well by the engineering community. We noted that a difference between RS method and MSVR method is that the former is fundamentally intended to solve inverse mathematical problems, while the latter transforms inverse problems into direct problems. Specifically, for the RS method to work, surrogate polynomial regression models must be established by first treating the design parameters and the structural properties as the inputs and the outputs, respectively. Then, optimizations must be undertaken to identify the structural parameters (the inputs) based on the structural properties (the outputs) measured on location. Inverse mathematical problems are solved in this process, and the accuracy of the identified structural parameters largely depends on the effectiveness of the optimization algorithms employed. In recent years, many advanced optimization algorithms that have been proven to be effective in practical use have been introduced into damage detection or the update of surrogate-model-based FE models [11–16]; however, practicing engineers generally utilize the optimization toolbox embedded in the numerical software without developing an in-house optimization code, and the effectiveness of these optimization toolboxes is questionable according to our experience. On the contrary, the MSVR method regards the structural properties and the design parameters as the inputs and the outputs, respectively, when establishing the surrogate models, and the targeted design parameters are directly obtained by inputting the measured structural properties into the established surrogate models. This is basically a process of solving direct problems which avoids utilizing the optimization algorithms to identify the structural parameters which generally provide inaccurate results, e.g., the local optimum. In this regard, the MSVR method should be technically more reliable than the RS method in updating FE models based on surrogate models. However, this contention still requires substantial validation through practical applications, a topic of the present study.

On the other hand, since steel bridges constantly undertake cyclic vehicle loads, fatigue becomes one of the most common failure modes of these bridges. Therefore, the assessment of the fatigue damage and the prediction of the fatigue life need to be embraced for the SHM of large steel bridges. Using the established, detailed FE models, the scenario of vehicles passing over the bridge can be simulated, and the accurate stress time histories at local positions can be obtained for use. However, an effective method is required to achieve fatigue analyses. The traditional fatigue analysis method is based on fatigue load spectrums of constant amplitudes [17–21]. A key limitation of this method is its inability to support continuous data acquisition. This can lead to inaccurate analyses as the significant effects of the loading sequence are neglected. To this end, Wei [22] has established an innovative high-circle fatigue damage accumulation theory based on the stress amplitude increments and has formulated a method for engineering applications based on the theory. Wei's fatigue analysis method overcomes the deficiencies of the traditional fatigue analysis method and has been proven to be effective by physical experiments [22]. However, this method has not been adopted in real-world SHM systems at this time. The new method needs to be incorporated into real-world engineering applications to validate its applicability.

In view of the above, validated, detailed numerical models and new effective fatigue analysis methods are crucial for SHM-oriented condition assessments of bridges. However, limited SHM systems have been geared with them so far. This article presents a novel process of establishing the SHM-oriented FE model and the subsequent fatigue analyses for Yingzhou Bridge. In 2009, Yingzhou Bridge, a specially shaped steel arch bridge with a main span of 120 m, was constructed in Luoyang, China. Before it was opened to traffic,

an SHM system was arranged on the bridge. The aforementioned simulation strategy, model-updating techniques, and fatigue analysis method were employed in the process of establishing the SHM system. We admit that the idea of using a surrogate model or meta-model to replace the finite element model for damage identification has been well-practiced. However, most of the studies performed were based on the backgrounds of small aviation and mechanical structures. On the contrary, Yingzhou Bridge is a very large system. Similar works that applied the existing numerical simulation and model-updating ideas to structures as complex as Yingzhou Bridge were rarely reported before. However, they are of significant practical importance on the safety assessments of large civil structures.

## 2. Engineering Background and Field Modal Test

Yingzhou Bridge is a half-through, tied arch bridge with a main span of 120 m. As shown in Figure 1, its specially shaped arch–rib system of reversed triangular cross-sections comprises one concrete-filled steel tubular arch in the middle and two hollow steel tubular subarches (one on each side). These subarches intersect at the skewbacks and are connected by lateral and inclined steel struts above. The steel hangers are uniformly attached to the three arches to form one vertical hanger plane and two inclined hanger planes. Suspended by hangers, the main deck is a steel composite box girder with prestressed cable ties inside. At the skewbacks, two huge concrete rigid triangles are designed which comprise caps, arch–ribs, and piers.

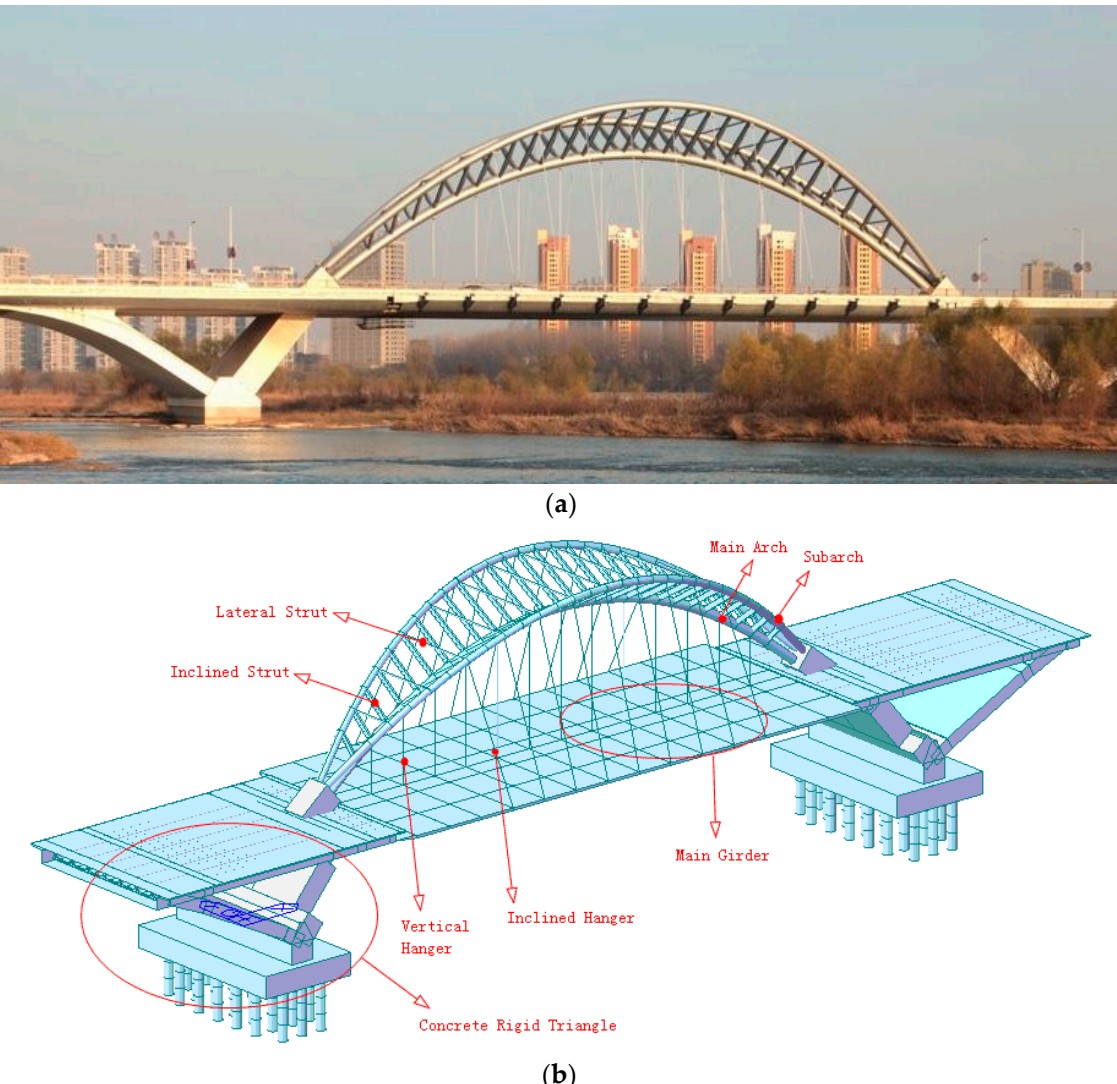

(**a**)

(**b**)

**Figure 1.** View and 3D sketch of Yingzhou Bridge. (**a**) Real view, (**b**) 3D sketch.

In 2009, a field modal test was undertaken on Yingzhou Bridge [23] that employed ambient excitations. The test utilized 46 moving measuring points that were uniformly distributed on the deck. The measured modal frequencies and the measured modal damping ratios are listed in Table 1. According to Table 1, the 1st~4th measured modal damping ratios fall into a proper range [0.5%, 4.5%], suggesting the good accuracy of the field modal test. According to the field modal test, the 1st~4th modal shapes for Yingzhou Bridge are: the vertical symmetric bending of the deck, torsion of the deck (1st), the vertical antisymmetric bending of the deck, and the torsion of the deck (2nd), respectively.

**Table 1.** Measured modal parameters and computed parameters for the initial bridge model.

| Mode No. | Mode Shape | Measured Modal Parameters | | Computed Modal Frequencies for the Initial Model | |
|---|---|---|---|---|---|
| | | Frequency (Hz) | Damping Ratio (%) | Result (Hz) | Difference (%) |
| 1 | Vertical symmetric bending of the deck | 1.25 | 4.3 | 0.677 | 45.84 |
| 2 | Torsion of the deck (1st) | 2 | 1.23 | 1.406 | 29.70 |
| 3 | Vertical antisymmetric bending of the deck | 2.88 | 0.71 | 1.912 | 33.61 |
| 4 | Torsion of the deck (2nd) | 3.88 | 1.56 | 2.378 | 38.71 |
| rms | | | | | 37.46 |

## 3. FE Simulation of Yingzhou Bridge

According to Cheng et al. [24], the whole structure can be divided into two load-carrying systems (see Figure 2). The first load-carrying system comprises the arch–rib system, two concrete rigid triangles, and the pre-stressed cable ties, which look like bows and carry the loads from the hangers. The second load-carrying system is composed of the hangers and the main deck, which mainly carries the vehicle loads [24].

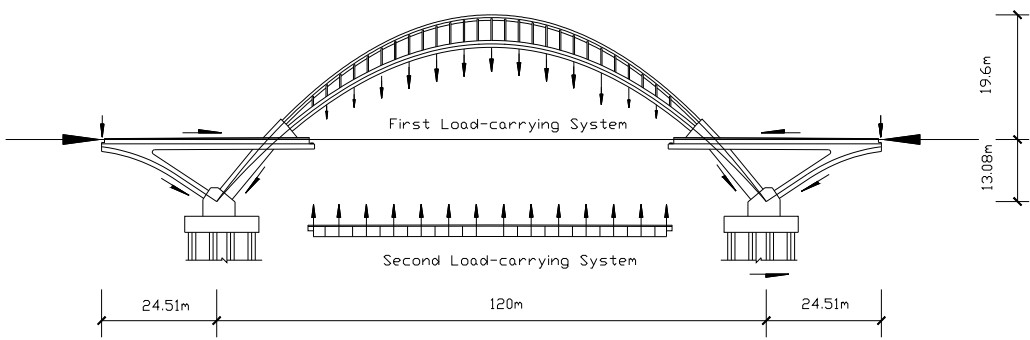

**Figure 2.** Load-transferring path for Yingzhou Bridge.

### 3.1. Modeling of the First Load-Carrying System

For the first system, shown in Figure 3, the specially shaped arch–rib system of Yingzhou Bridge was discretized into 169 linear, 3D beam elements. The cross-sectional shapes of these elements were in accordance with the design drawings, and perfect elasto-plastic material constitutive models were adopted for establishing these elements.

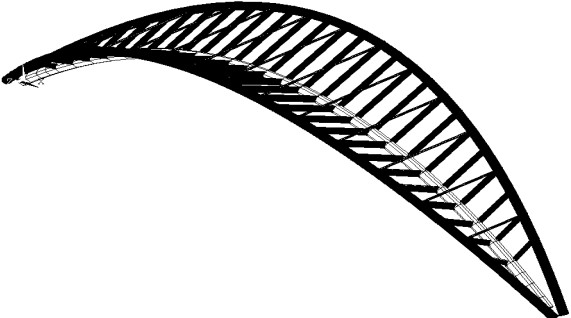

**Figure 3.** Model of the specially shaped arch–rib system.

The two concrete, rigid triangles have a complicated, irregular geometric shape. Thus, they were modeled using tetrahedron solid elements (Figure 4). As can be seen in Figure 4, local mesh refining was undertaken at the abutment since it was found by Sun [25] that the stress changes significantly at that position when the bridge is subjected to multiple design loads.

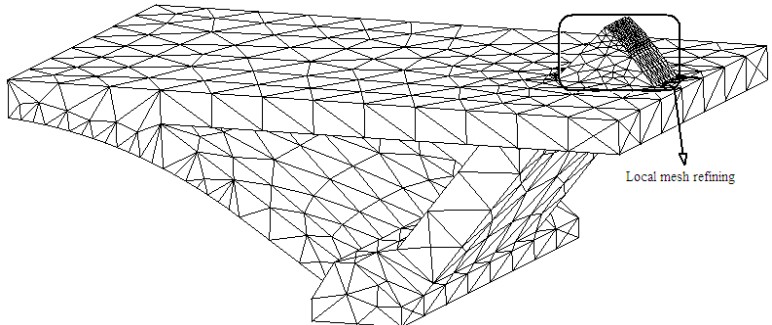

**Figure 4.** Model of a concrete rigid triangle.

Based on the multi-point constraining (MPC) technique [26], the model of the specially shaped arch–rib system, the models of the two concrete, rigid triangles, and the pre-stressed cable ties (bar elements) were assembled to form the first load-carrying system (Figure 5).

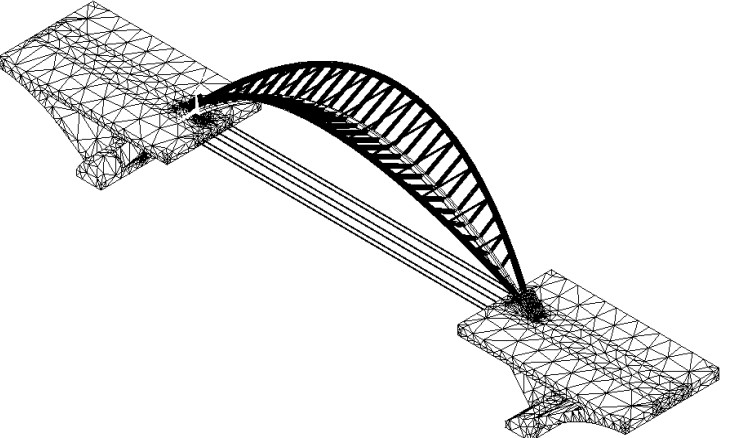

**Figure 5.** Model of the first load-carrying system.

### 3.2. Modeling of the Second Load-Carrying System

For the second system, the main deck of Yingzhou Bridge was modeled with finer details using orthotropic shell elements. As shown in Figure 6, all the details of the main deck were embraced for the deck model. In addition, the hangers were modeled by bar elements.

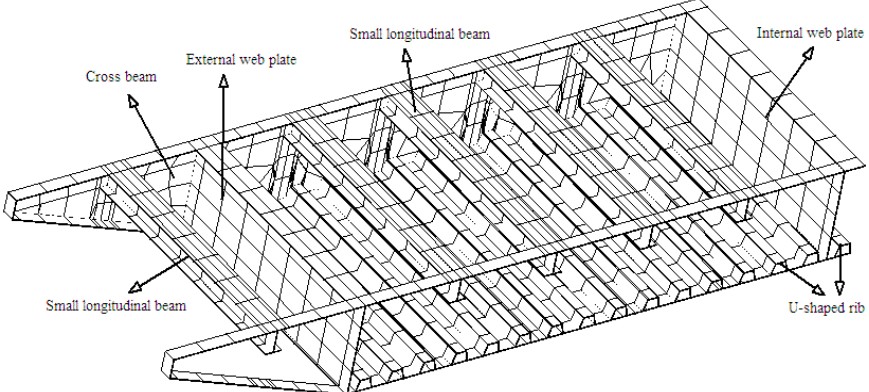

**Figure 6.** Detailed model of the main deck.

### 3.3. Preliminary Assemblage of the Bridge Model

As a traditional modeling practice [23,24,27], the first load-carrying system and the second load-carrying system were initially assembled to form the entire bridge using the coupling technique provided by ANSYS [26] (Figure 7). According to ANSYS [26], if the two nodes are coupled, a rigid connection can be established between two objects in the model. This may be different from the actual case of Yingzhou Bridge.

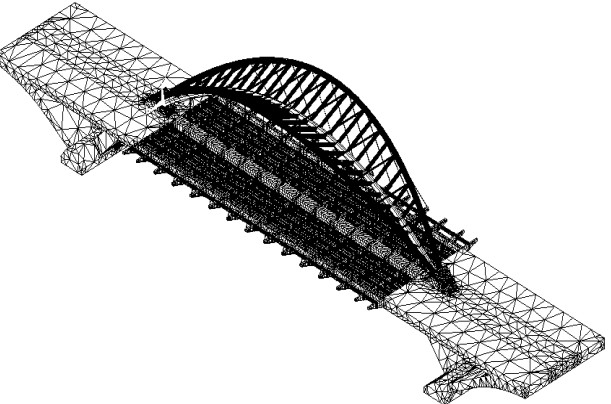

**Figure 7.** Model of whole Yingzhou Bridge.

## 4. Modal Analysis for Yingzhou Bridge

### 4.1. Preliminary Modal Analysis

After the FE model of Yingzhou Bridge was established, a modal analysis was undertaken. The calculated modal frequencies are shown in Table 1. As can be seen in Table 1, the 1st~4th modal frequencies that were calculated for the initial FE model are significantly smaller than the measured ones. The root mean square (rms) difference between the measured and the calculated modal frequencies reaches 37.46% for the initial FE model. In addition, the low-order modes calculated for the initial FE model (Figure 8) do not agree with the measured modes. As can be seen, the analytical modes presented in Figure 8 feature the vibrations of the entire bridge rather than vibrations of the deck alone as measured by the field modal test (see Section 2). Therefore, the initial model needs to be updated to improve its accuracy.

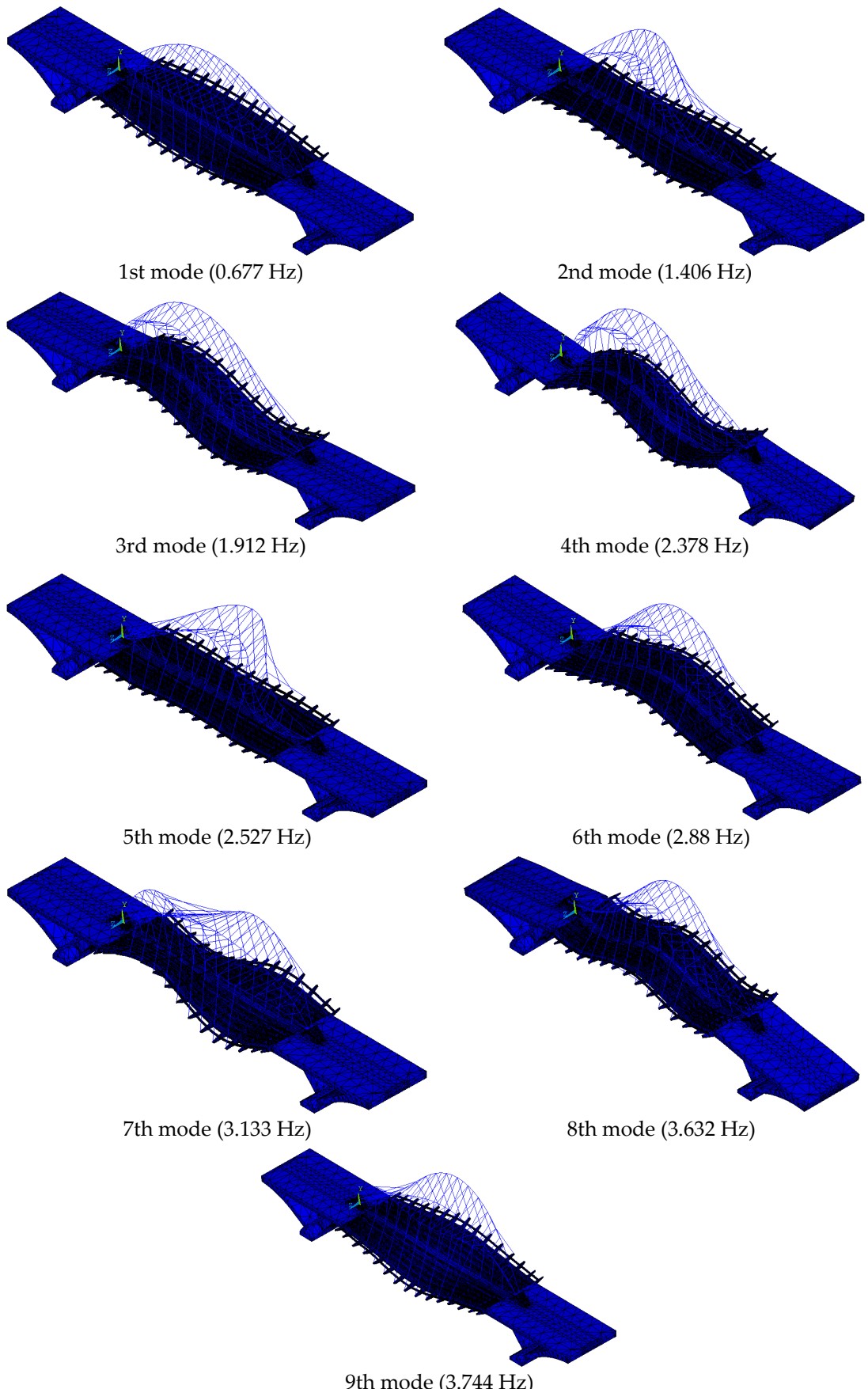

**Figure 8.** 1st~9th analytical modes for initial FE model.

### 4.2. Manual Tuning and Model Updating

According to Fei et al. [28], in comparison to the initial values of the parameters, the configurations have a much more significant influence on model updating. If there are erroneous configurations, the model may contain a structure error which can be difficult to correct by updating the model. The updated model may be nothing but a partially equivalent model. Therefore, the initial model has a pronounced influence on the success of model updating.

With regard to Yingzhou Bridge, it is noted that the first load-carrying system and the second load-carrying system are not connected rigidly. As shown in Figure 9, the bridge deck sits on lead rubber bearings that are arranged on the concrete rigid triangles. Thus, the two load-carrying systems are actually connected by elastic supports with an unknown stiffness. As suggested by the modal analysis results presented in Figure 8, they may not vibrate in a coordinated way.

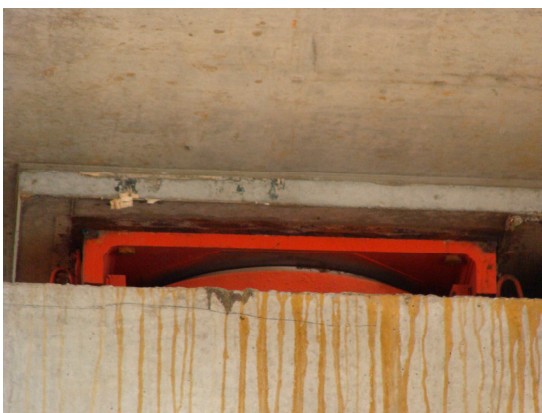

**Figure 9.** A lead rubber bearing on Yingzhou Bridge.

Based on the above, the deck model of Yingzhou Bridge was separated from the initial model, whose supports (lead rubber bearings and suspenders) were modeled by spring elements with fixed ends. A model updating strategy was formulated for the initial model of Yingzhou Bridge. The strategy consisted of three main steps. First, the stiffness of the lead rubber bearings supporting the deck model were treated as uncertain parameters, and parameter identification was undertaken based on the field modal test results. Since the detailed model is very complicated, two surrogate-model-based optimization methods (RS and MSVR) were utilized. Second, by comparing the results of the two surrogate-model-based parameter identification methods, the uncertain parameters could be determined. Finally, an updated model was obtained by correctly assembling the first load-carrying system and the second load-carrying system. The processes of parameter identification using the RS method and the MSVR method are presented in Sections 4.2.1 and 4.2.2, respectively.

#### 4.2.1. Model Updating Based on RS Method

Based on the theories of the RS method presented in Appendix A, the horizontal stiffness and the vertical stiffness of the bridge deck supports were regarded as uncertain parameters (parameter A and parameter B, respectively). Based on the design drawings, the initial values for the two parameters are 73,333.4 KN/m and 199,600,588.6 KN/m, respectively. Their allowable bound was ±50%. Using a central composite design (CCD) (*a* equals 1.68179), nine runs of sampling were undertaken (see Table 2). For each run, the modal analysis was conducted on the bridge deck model with the parameter values listed in Table 2 assigned to the horizontal stiffness and the vertical stiffness of its supports. The 1st~4th natural frequencies of the bridge deck were calculated and are shown in Table 2; these were used as the responses of the surrogate model. It must be stressed that we chose the CCD experimental design to generate samples for constructing the surrogate models

because Ref. [29] suggests that the quality of the model updated using the CCD sampling method is higher than the quality of model obtained with other sampling techniques.

**Table 2.** Samples based on CCD.

| | Parameter | | Response | | | |
|---|---|---|---|---|---|---|
| Run | A: Horizontal Stiffness of the Supports (KN/m) | B: Vertical Stiffness of the Supports (KN/m) | First Natural Frequency (Hz) | Second Natural Frequency (Hz) | Third Natural Frequency (Hz) | Fourth Natural Frequency (Hz) |
| 1 | 73,333.4 | 199,600,588.6 | 1.454 | 2.167 | 2.779 | 3.568 |
| 2 | 21,478.86 | 199,600,588.6 | 1.453 | 2.311 | 2.774 | 3.695 |
| 3 | 125,187.9 | 199,600,588.6 | 1.455 | 2.266 | 2.76 | 3.642 |
| 4 | 73,333.4 | 340,739,518.3 | 1.454 | 2.167 | 2.779 | 3.568 |
| 5 | 36,666.7 | 299,400,882.9 | 1.454 | 2.322 | 2.774 | 3.717 |
| 6 | 110,000.1 | 99,800,294.3 | 1.455 | 2.257 | 2.744 | 3.634 |
| 7 | 110,000.1 | 299,400,882.9 | 1.455 | 2.257 | 2.744 | 3.634 |
| 8 | 36,666.7 | 99,800,294.3 | 1.454 | 2.322 | 2.774 | 3.717 |
| 9 | 73,333.4 | 58,461,658.87 | 1.454 | 2.167 | 2.779 | 3.568 |

To establish the RS models, software Design-Expert (Version 6.0.10) was utilized. First, all data listed in Table 2 were input into the software. Four preliminary RS models that corresponded to the 1st~4th natural frequencies were established. These models adopted a full quadratic polynomial form. The process employed the least square method to estimate all of the relevant coefficients. The significances for all of the terms in the original models were examined using the F-test. The results for the models of the 1st~4th natural frequencies are shown in Table 3. The function forms of the RS models were modified based on these values. As can be seen in the table, the Prob > F values for some terms were large, indicating they had a relatively insignificant influence on the response of the structure and could be abandoned.

**Table 3.** Significance test results for terms in the original/modified regression models of the 1st~4th natural frequencies (significance level 5%).

| | Original Regression Model | | | | | | | |
|---|---|---|---|---|---|---|---|---|
| Term | First Mode | | Second Mode | | Third Mode | | Fourth Mode | |
| | F Value | Prob > F | F Value | Prob > F | F Value | Prob > F | F Value | Prob > F |
| A | 34.82 | 0.0006 | 4.01 | 0.0852 | 9.38 | 0.0183 | 6.69 | 0.0361 |
| B | 0.00 | 1.0000 | 0.00 | 1.0000 | 0.00 | 1.0000 | 0.00 | 1.0000 |
| $A^2$ | 1.30 | 0.2919 | 34.57 | 0.0006 | 7.40 | 0.0298 | 26.74 | 0.0013 |
| $B^2$ | 1.30 | 0.2919 | 1.42 | 0.2723 | 1.00 | 0.3497 | 1.31 | 0.2893 |
| AB | 0.00 | 1.0000 | 0.00 | 1.0000 | 0.00 | 1.0000 | 0.00 | 1.0000 |
| | Modified Regression Model | | | | | | | |
| Term | First Mode | | Second Mode | | Third Mode | | Fourth Mode | |
| | F Value | Prob > F | F Value | Prob > F | F Value | Prob > F | F Value | Prob > F |
| A | 44.77 | <0.0001 | 5.16 | 0.0493 | 12.06 | 0.0070 | 8.60 | 0.0167 |
| B | - | - | - | - | - | - | - | - |
| $A^2$ | 1.67 | 0.2285 | 44.45 | <0.0001 | 9.51 | 0.0131 | 34.38 | 0.0002 |
| $B^2$ | 1.67 | 0.2285 | 1.82 | 0.2097 | 1.29 | 0.2852 | 1.69 | 0.2260 |
| AB | - | - | - | - | - | - | - | - |

After the quadratic polynomial form modification, the RS regression models were rebuilt correspondingly. The significance test results for the modified regression models of the 1st~4th natural frequencies are listed in Table 4. According to Table 4, the new models can accurately represent the relationships between the factors and the responses. Taking the model of the 1st natural frequency as an example, there are three important results.

First, $F = 15.91 > F_{0.05}, P = 0.0006$. This shows that the regression model is significant and the lack of fit term is insignificant. Second, $R^2 = 0.8413$. This proves that the model is accurate and the experimental error is small. Third, the adequate precision (i.e., signal-to-noise ratio) equals 12.063. This proves that the signal is sufficient and the fitting is valid over the entire design space. Similar situations hold true for the other modified regression models (see Table 4).

**Table 4.** Significance test results for the modified regression models of 1st~4th natural frequencies.

| Regression Model | | Sum of Squares | DF | Mean Square | F Value | Prob > F | R² | Adequate Precision |
|---|---|---|---|---|---|---|---|---|
| First Mode | Model | $3.107 \times 10^{-6}$ | 3 | $1.036 \times 10^{-6}$ | 15.91 | 0.0006 | | |
| | Residual | $5.858 \times 10^{-7}$ | 9 | $6.509 \times 10^{-8}$ | | | 0.8413 | 12.063 |
| | Lack of fit | $5.858 \times 10^{-7}$ | 5 | $1.172 \times 10^{-7}$ | | | | |
| Second Mode | Model | 0.045 | 3 | 0.015 | 16.61 | 0.0005 | | |
| | Residual | $8.177 \times 10^{-3}$ | 9 | $9.085 \times 10^{-4}$ | | | 0.8470 | 11.161 |
| | Lack of fit | $8.177 \times 10^{-3}$ | 5 | $1.635 \times 10^{-3}$ | | | | |
| Third Mode | Model | $1.460 \times 10^{-3}$ | 3 | $4.867 \times 10^{-4}$ | 7.37 | 0.0085 | | |
| | Residual | $5.940 \times 10^{-4}$ | 9 | $6.600 \times 10^{-5}$ | | | 0.7108 | 7.346 |
| | Lack of fit | $5.940 \times 10^{-4}$ | 5 | $1.188 \times 10^{-4}$ | | | | |
| Fourth Mode | Model | 0.036 | 3 | 0.012 | 14.42 | 0.0009 | | |
| | Residual | $7.591 \times 10^{-3}$ | 9 | $8.435 \times 10^{-4}$ | | | 0.8278 | 10.659 |
| | Lack of fit | $7.591 \times 10^{-3}$ | 5 | $1.518 \times 10^{-3}$ | | | | |

Table 3 provides the significance test results for the terms in the modified regression models of the 1st~4th natural frequencies. After modification, all of the remaining terms had extremely significant influences on the responses. According to the modified polynomial form for all of the models, the interaction effect between the two factors can be neglected; however, the relationship between the factors and the response was not simply linear. The RS graph for the modified model of the 1st natural frequency is presented in Figure 10. Based on the results discussed above, the modified RS models can be used for parameter identification.

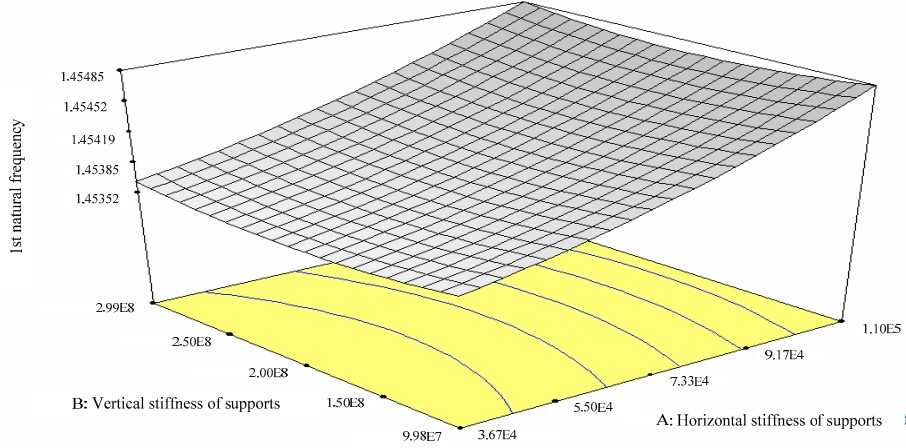

**Figure 10.** RS graph for the regression model of the 1st natural frequency (Hz) with parameters A (KN/m) and B (KN/m).

Finally, the optimization was performed on the four modified RS models with regard to the measured modal frequencies. The optimization toolbox embedded in software Design-Expert (Version 6.0.10) was used instead of the in-house optimization code developed, and the identified parameters were the horizontal stiffness of the supports (parameter A) = 73,333.40 KN/m and the vertical stiffness of the supports (parameter B) = 99,800,508.71 KN/m.

### 4.2.2. Model Updating Based on MSVR Method

Based on the theories of the MSVR method presented in Appendix B, the horizontal stiffness and the vertical stiffness of the bridge deck supports were regarded as uncertain parameters. The data listed in Table 2 were utilized to train the MSVR. The structure's natural frequencies were taken as the inputs of the trained MSVR, and the outputs of the MSVR were the stiffness of the supports. After the MSVR was trained, the measured modal frequencies were provided as inputs to the surrogate model. The identified parameters were the horizontal stiffness of the supports = 69,474 KN/m and the vertical stiffness of the supports = $1.891 \times 10^8$ KN/m. The horizontal stiffness of the supports identified by the MSVR method is close to the result of RS method; however, the vertical stiffness identified by the MSVR method is twice the value calculated using the RS method.

Modal analyses were conducted on the two bridge deck models that were updated using the two methods. The results are listed in Table 5. In Table 5, the rms difference between the measured and the calculated modal frequencies reduces from 37.46% for the initial model to approximately 10% for the two updated models. A comparison between the RS method and the MSVR method suggests that the results obtained using the two methods are similar; both the RS method and the MSVR method are effective with regard to the differences between the measured and the calculated modal frequencies.

**Table 5.** Measured modal frequencies and computed modal frequencies for the bridge deck with model updating.

| Mode No. | Mode Shape | Measured Results (Hz) | Updated Model Using the RS Method | | Updated Model Using the MSVR Method | |
|---|---|---|---|---|---|---|
| | | | Results (Hz) | Difference (%) | Results (Hz) | Difference (%) |
| 1 | Vertical symmetric bending of the deck | 1.25 | 1.454 | 16.32 | 1.454 | 16.32 |
| 2 | Torsion of the deck (1st) | 2 | 2.167 | 8.35 | 2.139 | 6.95 |
| 3 | Vertical antisymmetric bending of the deck | 2.88 | 2.779 | 3.51 | 2.778 | 3.54 |
| 4 | Torsion of the deck (2nd) | 3.88 | 3.568 | 8.04 | 3.546 | 8.61 |
| rms | | | | 10.16 | | 10.02 |

Finally, using the MPC technique, the model of the whole bridge was obtained by coupling the sides of the two concrete rigid triangles with the ends of spring elements, whose stiffnesses were identified by the RS method and the MSVR method, respectively. The updated models can facilitate the accurate SHM-oriented structural analyses.

### 4.3. Model Validation Using Mode Shapes

In addition to the natural frequencies, if more information measured on the physical truth (e.g., the mode shapes) is utilized for model updating, the space for tuning could be effectively compressed, and the model thereby becomes closer to the physical truth. However, compared with the modal frequencies, the mode shapes are harder to obtain

and less accurate, e.g., if one wants to measure the mode shapes of a large bridge, over ten accelerometers should usually be utilized; however, the bridge's modal frequencies can be measured by using only one accelerometer. Therefore, due to the limited field-sensing resources, most FE model updating practices reported by the literature up to this point were are only based on measured modal frequencies. To validate the effectiveness of the common practice, this article focuses on the natural frequencies. However, some mode shapes of Yingzhou Bridge have fortunately been measured on location and are reported in Refs. [23,30]. Therefore, we have additionally utilized this information to further validate the accuracies of the models that have already been updated using the RS and MSVR methods based on the natural frequencies as Ref. [1] suggests that the modeling and updating qualities can be further enhanced if the numerical model can demonstrate real dynamic characteristics other than those included in the updated procedure.

Figure 11 compares the 1st~4th mode shapes measured on location with those calculated from the two FE models that were updated using the RS method and the MSVR method, respectively. As can be seen, compared with the results calculated using the model updated according to the RS method, the mode shapes calculated using the model updated according to the MSVR method are closer to the field measurement data. For the model updated according to the RS method, the correlation coefficients between the physical test and the numerical data were 0.6214, 0.9497, 0.9695, and 0.9431 for 1st~4th mode shapes, respectively. However, they were 0.7748, 0.9553, 0.9593, and 0.9711, respectively, for the model updated according to the MSVR method. These suggest that the qualities of the model updated according to MSVR method are better than those of the model updated according to the RS method. Therefore, it can be inferred that the MSVR method is superior to the RS method in use for the surrogate-model-based model updating. Thereby, the model updated according to the MSVR method was utilized for the subsequent analyses.

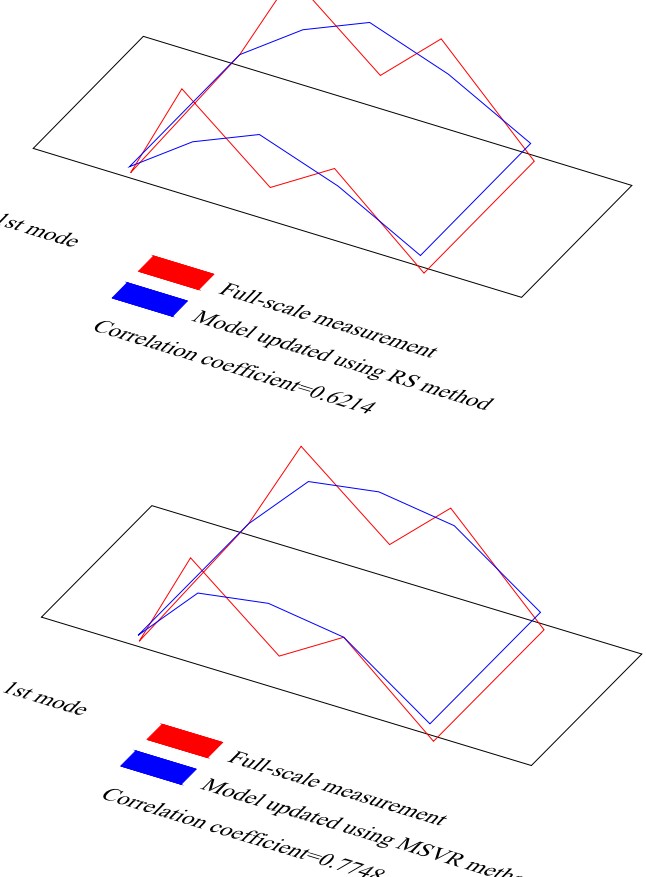

**Figure 11.** *Cont.*

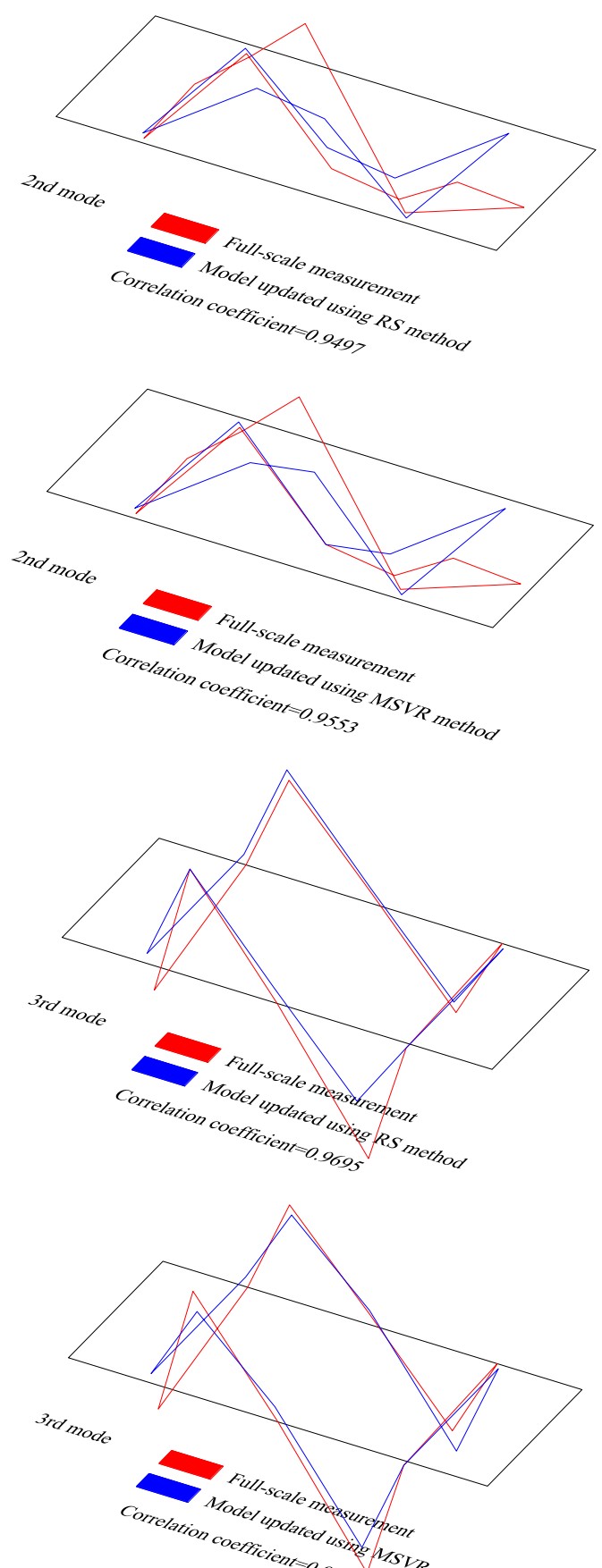

**Figure 11.** *Cont.*

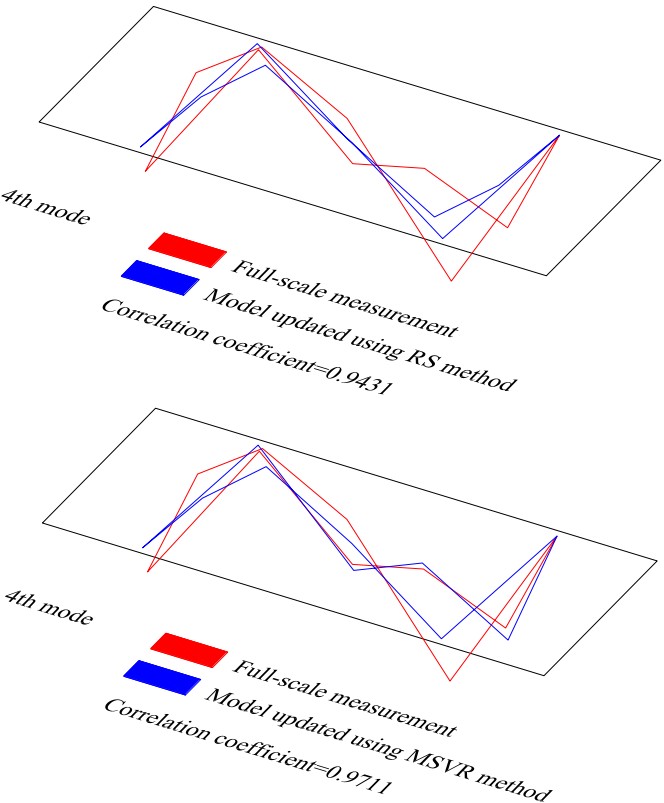

**Figure 11.** Comparisons of mode shapes obtained from field measurement and numerical analyses.

## 5. Vehicle-Induced Static Structural Responses

Before Yingzhou Bridge was opened for traffic, a field static load test was undertaken [23]. As is shown in Figure 12, four 300-KN trucks were used to evaluate the vehicle-induced static behavior of the bridge. Using the initial and the updated numerical models, the static load test was simulated. The results are presented in Figures 13–15. According to Figures 13 and 14, some differences can be observed in the vehicle-induced displacement response and the vehicle-induced stress response of the bridge deck between the initial and the updated models. In addition, Figure 15 suggests that the along-bridge stresses at the midspan cross section, which were calculated using the updated model, are closer to the data measured on the prototype [23] in comparison with the results calculated using the initial model. This proves the correctness of the model-updating approach.

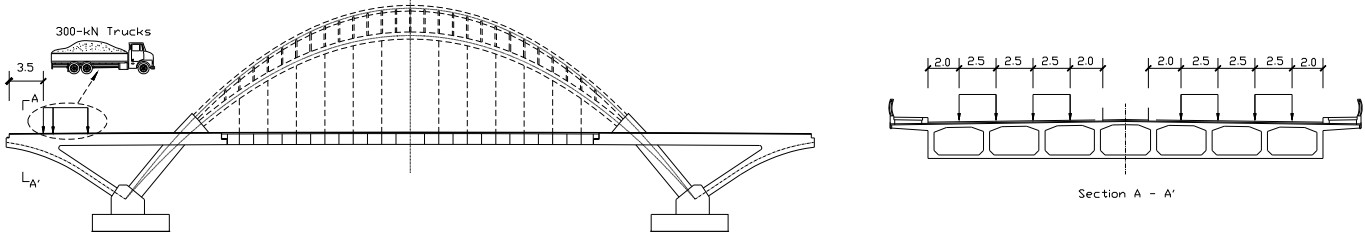

**Figure 12.** Loading positions for static load test (unit: m).

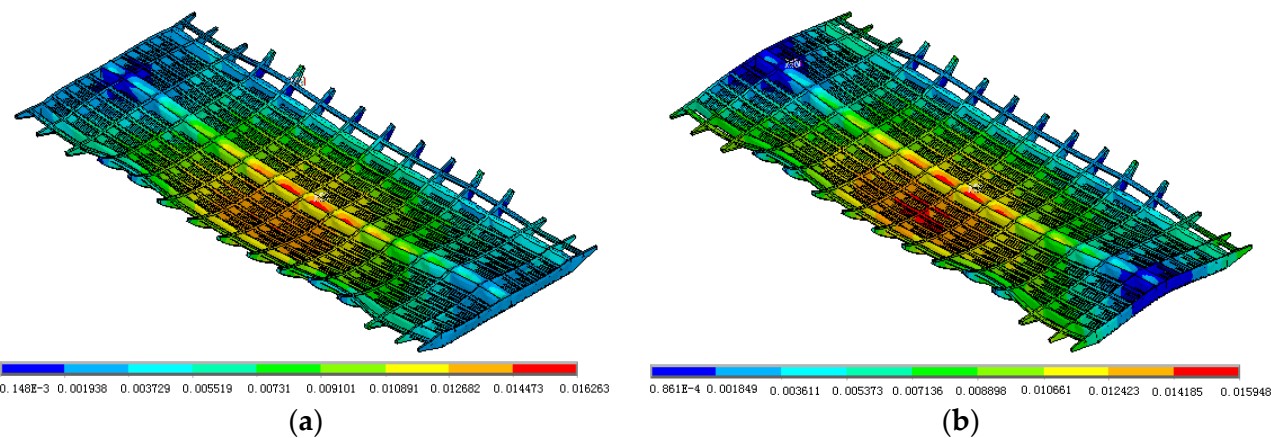

**Figure 13.** Contours of the calculated displacement vector sum of the bridge deck (unit: m). (**a**) Initial model, (**b**) Updated model.

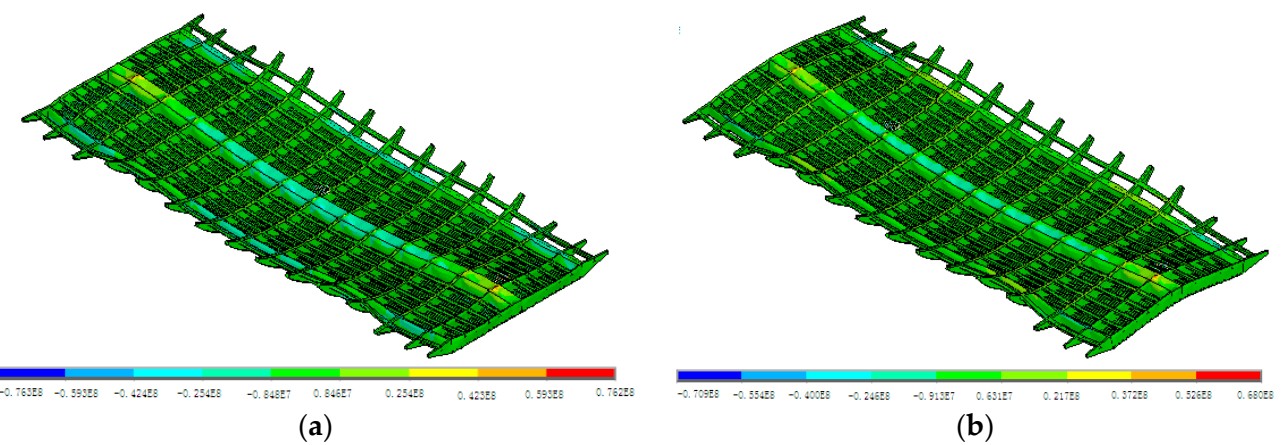

**Figure 14.** Contours of the calculated along-bridge stress on the bridge deck (unit: Pa). (**a**) Initial model, (**b**) Updated model.

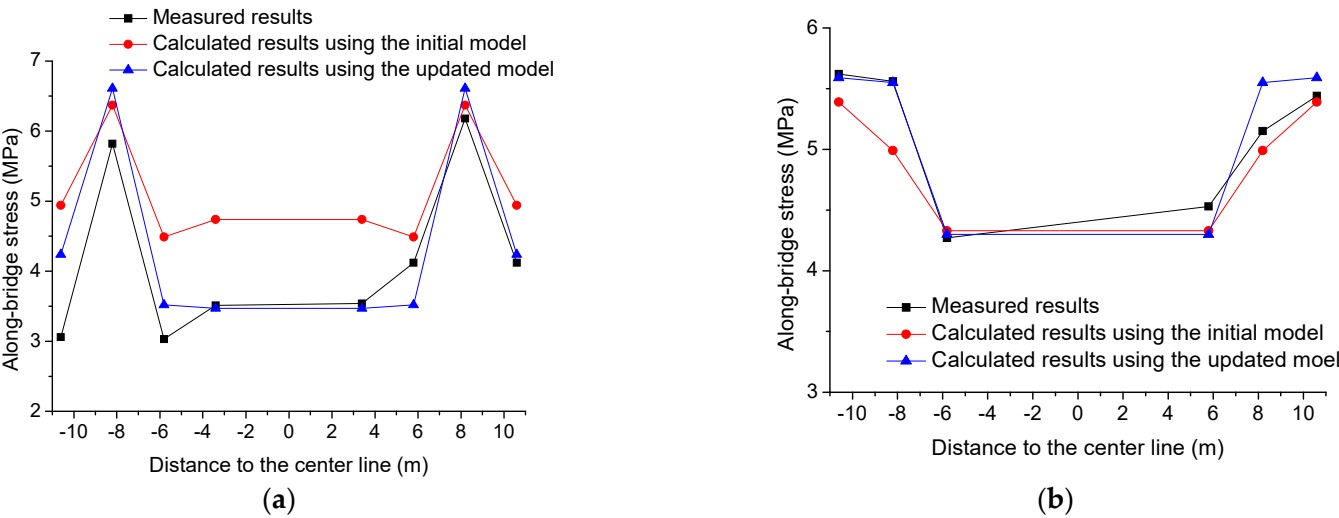

**Figure 15.** Measured and calculated along-bridge stresses at midspan cross section. (**a**) Stresses on top plate, (**b**) Stresses on base plate.

## 6. Vehicle-Induced Dynamic Structural Responses

When the SHM-oriented FE model was established, the fatigue analyses could be attempted. For fatigue analyses, the vehicle-induced local stress time-histories were obtained

through effective dynamic analyses using the Yingzhou Bridge's numerical model, which was updated in the former sections. A 300-kN truck moving along the bridge at the constant speed of 90 km/h was treated as a concentrated moving force. Transient analyses were performed by applying the concentrated moving force with the full solution method [26]. Stress stiffness effects and structural damping were considered for the dynamic analyses.

A vertical displacement time history calculated near the bearing for the whole process of the truck passing the steel box girder is shown in Figure 16. As can be seen, when the vehicle is close to the bearing (0–0.5 s), the displacement response is large; the displacement response is small when the vehicle moves away from the bearing (1–3.5 s). These observations agree with the basic rules of physics.

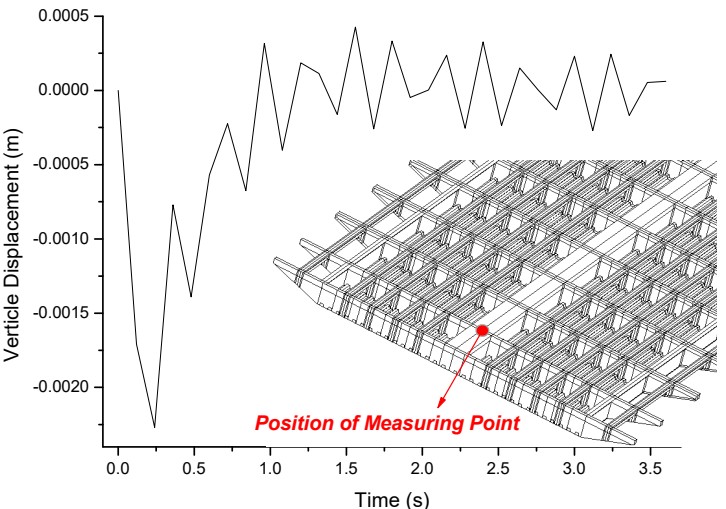

**Figure 16.** A vertical displacement time history obtained near the bearing.

An along-bridge local stress time-history calculated at the base-plate near the bearing is shown in Figure 17. As can be seen from Figure 17, the local stress vibrates in the range [−0.8 MPa, 1.2 MPa] during the whole vehicle-passing process. Obviously, the stress response induced by the vehicle is much smaller than the yield strength of the steel. However, as a typical cyclic stress, fatigue damages could possibly be induced by the local stress time-history presented in Figure 17.

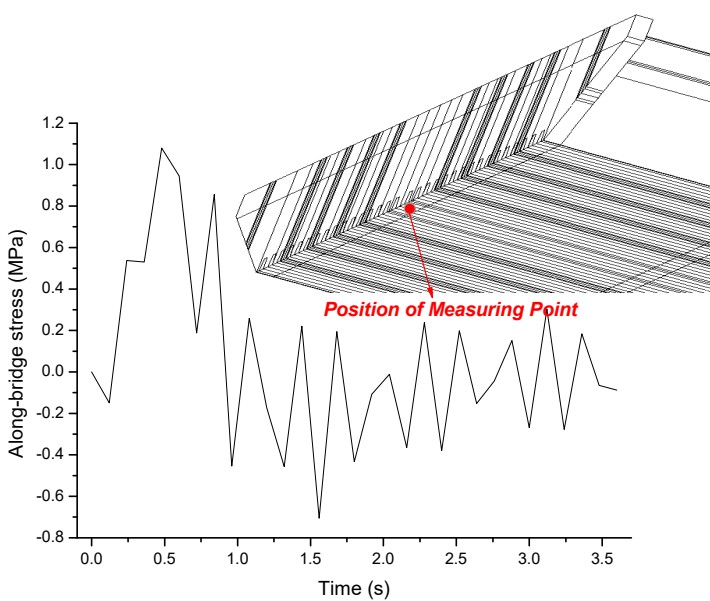

**Figure 17.** An along-bridge local stress time-history obtained at the base-plate near the bearing.

### 7. Fatigue Analyses

*7.1. High-Circle Fatigue Damage Accumulation Theory Proposed by Wei [22]*

The phenomenological theory in damage mechanics utilizes the concept of micro-plasticity to describe the deformation mechanism of metallic materials in the process of high-circle fatigue. The micro-plasticity can be interpreted as an inelastic deformation. This deformation is induced by the stress within the elastic range in the scenario of cyclic loads; the energy dissipations happen after a large magnitude of stress cycles. As with the usual plastic strain, the micro-plastic strain, $e_p$, can be formulated in the power function:

$$e_p = \left(\frac{\sigma}{K}\right)^{\beta} \tag{1}$$

where $\sigma$ is the stress and $\beta$ and $K$ are micro-plastic coefficients of the metallic material. Then, under the uniaxial alternating stress below the yield strength, the constitutive relation of the steel can be expressed as:

$$\varepsilon = \frac{\sigma}{E} + \left(\frac{\sigma}{K}\right)^{\beta} \tag{2}$$

where $\varepsilon$ is the strain and $E$ is the elastic modulus before damage. The damage variable, $D$, is defined as:

$$D = 1 - \frac{\widehat{E}}{E} \tag{3}$$

where $\widehat{E}$ is the elastic modulus after the damage. Based on the hypothesis of strain equivalence in the phenomenological theory, the constitutive relation, considering the damage, can be formulated as:

$$\varepsilon = \frac{\sigma}{E(1-D)} + \left(\frac{\sigma}{K}\right)^{\beta} \tag{4}$$

Extended this to the usual alternating stress case, Equation (4) can be written as:

$$\varepsilon_i = \frac{\sigma_i}{E(1-D)} + \left(\frac{\sigma_i}{K}\right)^{\beta} \tag{5}$$

$$\varepsilon_i = \frac{1}{\sqrt{2}(1+\mu)}\left[(\varepsilon_1 - \varepsilon_2)^2 + (\varepsilon_2 - \varepsilon_3)^2 + (\varepsilon_3 - \varepsilon_1)^2\right]^{1/2} \tag{6}$$

$$\sigma_i = \frac{1}{\sqrt{2}}\left[(\sigma_1 - \sigma_2)^2 + (\sigma_2 - \sigma_3)^2 + (\sigma_3 - \sigma_1)^2\right]^{1/2} \tag{7}$$

where $\varepsilon_1, \varepsilon_2, \varepsilon_3$ are the first, second, and third principal strains, respectively; and $\sigma_1, \sigma_2, \sigma_3$ are the first, second, and third principal stresses, respectively. Assuming that the free energy density function, $\phi^*$, is the function of the equivalent stress, the free energy density function can be expressed as:

$$\phi^* = \frac{\sigma_i{}^2}{2\rho E(1-D)} + \frac{\sigma_i^{\beta+1}}{\rho K^{\beta}(\beta+1)} \tag{8}$$

where $\rho$ is the density. By taking variation on Equation (8), the damage energy consumption rate, $\overline{Y}$, is obtained:

$$\overline{Y} = \rho\frac{\partial \phi^*}{\partial D} = \frac{\sigma_i{}^2}{2E(1-D)^2} \tag{9}$$

In the phenomenological theory, the dissipative potential function, $\Pi$, has orthogonal fluidity:

$$\frac{\dot{D}}{\frac{\partial \Pi}{\partial D}} = \frac{\dot{e}_{pj}}{\frac{\partial \Pi}{\partial e_{pj}}} \, (j = 1, 2, 3) \tag{10}$$

where $\dot{D}$ is the damage accumulation rate and $\dot{e}_{pj}$ is the micro-plastic strain rate in the corresponding direction. Assuming that the partial derivative of $\Pi$ with respect to the micro-plastic strain is the stress deviator in the corresponding direction, $S_j$, and the partial derivative of $\Pi$ with respect to the damage variable is the damage energy consumption rate:

$$\frac{\partial \Pi}{\partial e_{pj}} = S_j \tag{11}$$

$$\frac{\partial \Pi}{\partial D} = \overline{Y} \tag{12}$$

We can obtain:

$$\dot{D} = \frac{\overline{Y} \cdot \dot{e}_{pj}}{S_j} = \frac{3\overline{Y} \cdot \dot{e}_{pi}}{2\sigma_i} \tag{13}$$

where $\dot{e}_{pi}$ is the micro-plastic equivalent strain rate. Substituting Equation (9) and the expression of the micro-plastic equivalent strain rate into Equation (13), the relation between the damage accumulation rate and the equivalent stress growth rate can be established:

$$\dot{D} = \frac{3\beta\sigma_i^{\beta}}{4EK^{\beta}(1-D)^2}\dot{\sigma}_i \tag{14}$$

For its physical significance, Equation (14) can be rewritten as:

$$\dot{D} = \frac{3\beta\sigma_i^{\beta}}{4EK^{\beta}(1-D)^2}|\dot{\sigma}_i| \tag{15}$$

Assuming that the equivalent stress monotonically increases during the time interval $[t_0, t_1]$, the high-circle fatigue damage function can be obtained by solving Equation (15):

$$D_t = 1 - \sqrt[3]{(1-D_0)^3 - \frac{9\beta}{4EK^{\beta}(1+\beta)}(\sigma_{i,t}^{\beta+1} - \sigma_{i,0}^{\beta+1})} \tag{16}$$

where, $D_0$ and $\sigma_{i,0}$ are the damage accumulation variable and the equivalent stress at the time $t_0$, respectively; and $D_t$ and $\sigma_{i,t}$ are those at the time $t$ ($t \in [t_0, t_1]$).

### 7.2. Coefficient Determination for the High-Circle Fatigue Damage Function

For the high-circle fatigue damage function (Equation (16)) to be utilized, the coefficients $\beta$ and $K$ in Equation (16) need to be determined. According to [22], the coefficient determination can be realized by fitting the fatigue test data. The fatigue test data for two steel components reported by Wei [22], namely, the 4# and 10# components, are shown in Figure 18. The 4# component is manufactured by butt welding. The cyclic stress applied on the 4# component is $75 \pm 50$ MPa. According to Figure 18a, the fatigue life for the 4# component is $1 \times 10^6$ cycles, and the ultimate damage D is 0.089. With respect to the 10# component, it is manufactured by T-type welding. The cyclic stress acting on the 10# component is $100 \pm 75$ MPa. According to Figure 18b, the fatigue life for the 10# component is also $1 \times 10^6$ cycles, and the ultimate damage D is 0.066.

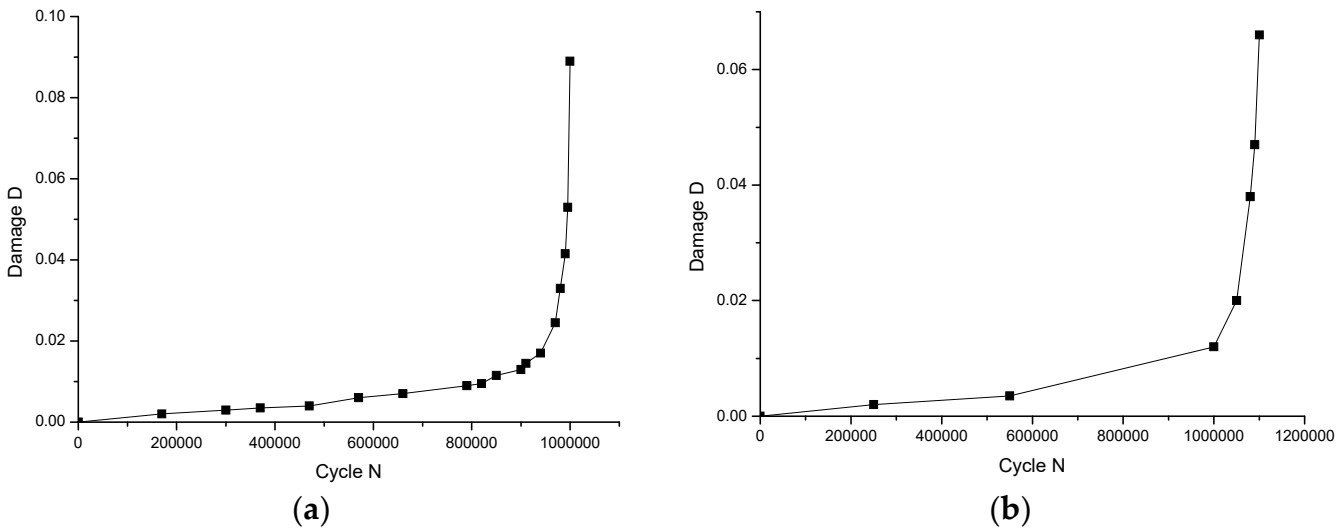

**Figure 18.** Fatigue damage cyclic loading number for steel components (reproduced from Ref. [22]). (**a**) 4# component, (**b**) 10# component.

According to [22], under uniaxial constant-amplitude cyclic loads $[\sigma_m - \sigma_a, \sigma_m + \sigma_a]$, a steel component's fatigue life can be estimated according to:

$$N_{CR}^{\sigma_m} = \frac{2EK^\beta(\beta+1)[1-(1-D)^3]}{9\beta[(\sigma_m+\sigma_a)^{\beta+1}-(\sigma_m-\sigma_a)^{\beta+1}]} \tag{17}$$

By substituting the data for the 4# and 10# components into Equation (17), two equations with two unknowns are obtained. By solving the equations, $\beta \approx 0.1$ and $K \approx 1$ are calculated.

### 7.3. Fatigue Damage Accumulation on Yingzhou Bridge Induced by a Passing Vehicle

Based on Sections 7.1 and 7.2, the following high-circle fatigue damage function can be obtained to calculate the fatigue damage accumulation on the newly built Yingzhou Bridge:

$$D_t = 1 - \sqrt[3]{1 - (9.93e - 7) * (\sigma_{i,t}^{1.1} - \sigma_{i,0}^{1.1})} \tag{18}$$

Using the data presented in Figure 17, the fatigue damage accumulation at the base-plate near the bearing of the Yingzhou Bridge was calculated for a 300-kN truck passing the bridge to be $D = 4.75e - 11$. From this, it can be concluded that severe fatigue damage is not likely to be induced by normal moving vehicles.

### 8. Conclusions

The main findings of this study concerning the SHM-oriented FE modeling and the fatigue analyses for Yingzhou Bridge are summarized below:

(1) To analyze the local structural behavior (e.g., fatigue), all the structure's components are simulated with a finer detail for SHM-oriented structural models. As this leads to models with a high complexity, surrogate-model-based methods can be employed to update the model. The present study suggests that the MSVR method is superior to the RS method in surrogate-model-based model updating with regard to the efficiency and the effectiveness of the updated models in reproducing the mode shapes of the physical truth. The correlation coefficients between the physical test and the numerical data are 0.6214, 0.9497, 0.9695, and 0.9431 for the 1st~4th mode shapes for the model updated according to RS method. However, they are 0.7748, 0.9553, 0.9593, and 0.9711, respectively, for the model updated according to the MSVR method. Although some research has demonstrated the efficiency and effectiveness of the RS method in bridge model updating, we cannot recommend this model updating approach for use based

on the present case study. An explanation for the weak point of RS method is that RS method is fundamentally intended to solve inverse mathematical problems, and effective optimization algorithms are required. Since practicing engineers generally utilize the optimization toolbox embedded in the numerical software, the effectiveness of which is questionable, the usual model updating practice utilizing RS method cannot guarantee the accuracy of the updated model;

(2) Some model updating focuses on the model parameter error and disregards the model structure error. However, if there are erroneous configurations, the model is difficult to correct through parameter identification and manual tuning is required. With regard to the bridges' numerical models, the simulation of the connections between the main girder and the supports deserves attention for its correctness;

(3) The new fatigue analysis method based on the high-circle fatigue damage accumulation theory proposed by Wei [22] takes into account the significant effects of the loading sequence. Therefore, the accuracy of the new method exceeds the accuracy of the traditional method based on rainflow counting and the Palmgren–Miner rule. By employing the new method to analyze the vehicle-induced fatigue damage on Yingzhou Bridge, the applicability of the new method to a real-world engineering case is validated in the present study. Using the new method, the fatigue damage accumulation at the base-plate near the bearing of the Yingzhou Bridge for a 300-kN truck passing the bridge was calculated to be $D = 4.75\mathrm{e} - 11$. From this, it can be concluded that severe fatigue damage is not likely to be induced by normal moving vehicles. Finally, it should be admitted that although the new fatigue analysis method is proven to be effective in dealing with the vehicle-induced fatigue analysis of a steel arch bridge in this article, substantial validations are still required to prove its effectiveness in dealing with other structures subjected to other in-service actions or extreme events. Our future works will focus on this topic.

**Author Contributions:** Conceptualization, X.-X.C.; methodology, X.-X.C. and M.-D.C.; software, X.-X.C.; validation, L.D. and X.-X.C.; formal analysis, X.-X.C.; investigation, X.-X.C. and M.-D.C.; resources, X.-X.C.; data curation, X.-X.C. and M.-D.C.; writing—original draft preparation, L.D.; writing—review and editing, X.-X.C.; visualization, L.D.; supervision, X.-X.C.; project administration, X.-X.C.; funding acquisition, X.-X.C. All authors have read and agreed to the published version of the manuscript.

**Funding:** This research and the APC were funded by the Key Science and Technology Project of Jiangxi Provincial Department of Communications [grant number 2019Z0002].

**Data Availability Statement:** The data presented in this study are available on request from the corresponding author.

**Acknowledgments:** The authors gratefully acknowledge the financial supports from the Key Science and Technology Project of Jiangxi Provincial Department of Communications (Grant No. 2019Z0002).

**Conflicts of Interest:** The authors declare that there are no conflict of interest regarding the publication of this article.

**Appendix A. RS Method**

The RS method integrates the experimental design technique with mathematical statistics techniques. The four main steps of the RS method are as follows:

(1) Samples are collected based on a widely used experimental design method, including orthogonal design, central composite design (CCD), Box–Behnken design, and D-optima design. These experimental design methods ensure the accuracy of the RS models with limited sampling. The CCD is a common method which leads to three categories of experiments, i.e., corner point experiments based on orthogonal table using the maximum and the minimum levels for each factor, star point experiments (star points have coded coordinates $(\pm a, 0, \ldots, 0)$, $(0, \pm a, \ldots, 0)$, etc., with one coordinate being $\pm a$ and all other

coordinates being zeros), and a center point experiment with coordinates $(0, 0, \ldots, 0)$. The CCD method is adopted in this paper.

(2) Parameter screening is conducted based on variance analysis. The basic idea of variance analysis is to divide the total square error of sampling characteristics into two parts. Significant parameters can be distinguished by a hypothesis test, the F-test. For parameter $A$, supposing that

$$F_A = \frac{S_A / f_A}{S_e / f_e} \sim F(f_A, f_e) \tag{A1}$$

$S_A$ and $S_e$ are square errors caused by uncertain parameters and by experiments, respectively, and $f_A$ and $f_e$ are the dimensions of freedom of $S_A$ and $S_e$, respectively. For a given significance level $\alpha$, if $F_A \geq F_{1-\alpha}(f_A, f_e)$ or the probability $(F > F_A)$ is small, the parameter is significant. Based on the results of the F-test, significant parameters are selected and the function forms of the RSs are correspondingly modified.

(3) RSs are fitted employing a quadratic polynomial as the response function. Supposing the response of the structure is $y$, and $x_i, i = 1, 2, \ldots, k$ are the significant uncertain parameters, the form of response function is:

$$y = \beta_0 + \sum_{i=1}^{k} \beta_i x_i + \sum_i \sum_j \beta_{ij} x_i x_j + \sum_{i=1}^{k} \beta_{ii} x_i^2 \tag{A2}$$

where $x_i \in \left[x_i^l, x_i^u\right]$, $x_i^l, x_i^u$ are the lower and upper bounds of the design parameter $x_i$. With the samples $y_1, y_2, \ldots, y_n$ collected in the first step, RSs are fitted using the least square method. Therefore, the unknown coefficients in Equation (A2), $\beta_0, \beta_i, \beta_{ij}$ and $\beta_{ii}$ are estimated. The following indicator can be used to check the accuracy of RS:

$$R^2 = 1 - \frac{\sum\limits_{j=1}^{N} \left[y_{RS}(j) - y(j)\right]^2}{\sum\limits_{j=1}^{N} \left[y(j) - \overline{y}\right]^2} \tag{A3}$$

where $y_{RS}(j)$ and $y$ are the outputs of the RS and the experiment, respectively; and $\overline{y}$ is the mean of $y$. The greater the value of $R^2$, the more accurate the RS model is. If the accuracy of the RS is not satisfactory, the experimental design method or the function form can be modified.

(4) Parameter optimization is conducted. The optimization can be summarized as the following mathematical problem:

$$Min\|R(p)\|_2^2, R(p) = \{f_A(p)\} \tag{A4}$$

$$st\,VLB \leq P \leq VUB$$

where $p$ is the design parameter; $\{f_A\}$ is the analytical structural response vector; $VLB$ and $VUB$ are the lower and upper bound of design space, respectively; and $R$ is the residual vector.

**Appendix B. MSVR Method**

The MSVR method is a new machine learning method based on statistical learning theory. Assuming that $x$ is the input variable, the regression function $F(x)$ can be expressed as

$$F(x) = \phi(x)^T W + B \tag{A5}$$

where $\phi(.)$ is the nonlinear mapping vector in the high-dimensional space and $W = [w^1, w^2, \ldots, w^N]$ and $B = [b^1, b^2, \ldots, b^N]$ are defined as N-dimensional linear regressors in the high-dimensional feature space.

According to the MSVR method, the structural risk minimization in multi-output cases is equal to the following constrained optimization problem:

$$\min L(W, B) = \frac{1}{2}\sum_{j=1}^{N}\left\|w^j\right\|^2 + C\sum_{i=1}^{L}L(u_i) \tag{A6}$$

where $L(u)$ is the loss function defined in the hyper-spherical zone expression as follows:

$$L(u) = \begin{cases} 0, u < \varepsilon \\ u^2 - 2u\varepsilon + \varepsilon^2, u > \varepsilon \end{cases} \tag{A7}$$

where $u_i = \|e_i\| = \sqrt{e_i^T e_i}$, $e_i^T = y_i^T - \phi^T(x_i)W - B$ and $\varepsilon$ is the hyper-spherical insensitive zone. When $\varepsilon = 0$, this problem reduces to an independent, regularized kernel least square regression for each component. However, for a nonzero $\varepsilon$, the solution takes into account all of the outputs to construct an individual regressor and is able to obtain the best global predictions.

According to Teng et al. [10], the main steps of model updating based on the MSVR method are as follows:

(1) Construct the samples. Samples of design parameters can be determined based on the CCD, and the responses of the structure can be obtained by FE analysis for each sample;

(2) Train the MSVR. The MSVR is trained with the responses and the corresponding design parameters, which are regarded as the inputs and the outputs, respectively;

(3) Update the design parameters. The measured responses are input to the trained MSVR; the outputs are the target design parameters.

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
