# Peer review of "Structural-Health-Monitoring-Oriented Finite Element Model for a Specially Shaped Steel Arch Bridge and Its Application"

_mca, doi:10.3390/mca28020033_

Round 1

Reviewer 1 Report

This paper presents a finite element model updating method assisted by a surrogate model for generating the high-fidelity numerical model using the measured modal information, i.e., natural frequencies. The paper uses an actual engineering structure, YingZhou Bridge as an example for validation, which is convincing. Nevertheless, there are still some significant issues to be addressed by the authors:

1. The novelty of the work is unclear. Finite element model updating is a prevailing method for structural health monitoring. To reduce the computational cost, a lot of studies have used the surrogate model or meta-model to replace the finite element model for damage identification. this idea is well-established and has been extensively practiced. It seems this paper extends a similar idea to the fatigue analysis. from an algorithmic aspect, authors are expected to clarify their contribution.

2. Introduction is not concrete.  As said previously, finite element modeling is a mainstream method for damage identification. However, the relevant discussion doesn’t involve the state-of-the-art. The authors have briefly mentioned the aspect of using meta-model to improve performance. The efforts made to enhance the optimization algorithms to advance the SHM capacity are totally lacking. Some recently published articles (published 2021 and 2022) in this regard need to be discussed to reflect the state of the art.

https://doi.org/10.1016/j.ymssp.2020.107121

https://doi.org/10.1007/s00366-021-01511-7

https://doi.org/10.1016/j.jsv.2021.116331

https://doi.org/10.1016/j.apm.2020.09.002

https://doi.org/10.1016/j.engstruct.2022.114312

https://doi.org/10.1177/1475921720926970

3. It seems the natural frequencies are of interest for model updating. Did authors ever consider involving the mode shape as well. Usually, the space for tuning is very large if limited information is used. Also, the model may not represent the actual structure if limited response information is used. It would be helpful to clarify this. I am interested to see what will be going on if the mode shapes are further utilized.

4. The details to establish the RS and MSVR models are missing. This is the critical step because of the trade-off between efficiency and accuracy. how to do the design of experiment (DOE) to generate samples for constructing the surrogate models in this research?

5. Authors mentioned “optimization” used in the model updating procedure. Did the authors just utilize the optimization toolbox in numerical software or develop the in-house optimization code? Those information needs to be added.

6. I would suggest enlarging the label size in Figures 10 and 14. The fonts are too small to read. In Figure 10, can authors use scientific notations instead of the real value?

7. The title of the work includes the keyword “fatigue analysis”. However, in section 7, when the authors talked about the fatigue analysis, the theory used all is from the other’s work and no specific result is given. The work cited is a Ph.D. dissertation dated back to 2009, which is quite old. This needs to be carefully explained. Otherwise, the contribution of the work becomes questionable, and the title somehow should be changed since the fatigue analysis actually is not conducted.

Author Response

Reviewer 1

This paper presents a finite element model updating method assisted by a surrogate model for generating the high-fidelity numerical model using the measured modal information, i.e., natural frequencies. The paper uses an actual engineering structure, YingZhou Bridge as an example for validation, which is convincing. Nevertheless, there are still some significant issues to be addressed by the authors:

  1. The novelty of the work is unclear. Finite element model updating is a prevailing method for structural health monitoring. To reduce the computational cost, a lot of studies have used the surrogate model or meta-model to replace the finite element model for damage identification. this idea is well-established and has been extensively practiced. It seems this paper extends a similar idea to the fatigue analysis. from an algorithmic aspect, authors are expected to clarify their contribution.

Response: Yes, the idea of using surrogate model or meta-model to replace the finite element model for damage identification has been well practiced. However, most of those performed studies are based on the engineering backgrounds of small aviation and mechanical structures, e.g. Refs. [R1-R3]. On the contrary, Yingzhou Bridge is a very large system. Similar works applying existing detailed numerical simulation and model updating ideas to structures as complex as Yingzhou Bridge were rarely reported before. However, they are of significant practical importance on safety assessments of large civil structures. We have clarified this contribution in the revised manuscript (see the part highlighted in blue). Thank you for your good suggestion.

References:

[R1] Fei Q, Zhang L, and Wang T. Evaluation of sampling techniques for neural networks applied to computational simulation of structures. Earthquake Eng Eng Vibration 2005; 25(1): 21–25, (in Chinese).

[R2] Fei Q, Zhang L, Li A, et al. Finite element model updating using statistics analysis. J Vibration Shock 2005; 24(3): 23–26, (in Chinese).

[R3] Fei Q, Li A, and Zhang L. Study on finite element model updating of nonlinear structures using neural networks. J Astronautics 2005; 26(3): 267–269, (in Chinese).

  1. Introduction is not concrete. As said previously, finite element modeling is a mainstream method for damage identification. However, the relevant discussion doesn’t involve the state-of-the-art. The authors have briefly mentioned the aspect of using meta-model to improve performance. The efforts made to enhance the optimization algorithms to advance the SHM capacity are totally lacking. Some recently published articles (published 2021 and 2022) in this regard need to be discussed to reflect the state of the art.

https://doi.org/10.1016/j.ymssp.2020.107121

https://doi.org/10.1007/s00366-021-01511-7

https://doi.org/10.1016/j.jsv.2021.116331

https://doi.org/10.1016/j.apm.2020.09.002

https://doi.org/10.1016/j.engstruct.2022.114312

https://doi.org/10.1177/1475921720926970

Response: First, thank you for recommending these good publications. We have read and cited all of them [21-26]. Secondly, as you indicated, the Introduction part is not concrete without the relevant discussions involving the state-of-the-art. Therefore, we have strengthened the Introduction part by theoretically analyzing the RS method and the MSVR method and comparing them in view of the effectiveness of both the traditional optimization algorithms utilized and the advanced optimization algorithms proposed in these two years [21-26] (see the part highlighted in red). We come to the conclusion that the MSVR method should be technically more reliable than the RS method in FE model updating based on surrogate models, and further suggest that this contention still requires substantial validations through practical applications.

  1. It seems the natural frequencies are of interest for model updating. Did authors ever consider involving the mode shape as well. Usually, the space for tuning is very large if limited information is used. Also, the model may not represent the actual structure if limited response information is used. It would be helpful to clarify this. I am interested to see what will be going on if the mode shapes are further utilized.

Response: Thank you for your good suggestion. Yes, if more information measured on the physical truth (e.g., the mode shapes) is utilized for model updating, the space for tuning could be effectively compressed, and the model thereby becomes closer to the physical truth. However, compared with the modal frequencies, the mode shapes are harder to obtain and less accurate, e.g., if one wants to measure a large bridge’s modes shapes, over ten accelerometers should usually be utilized; but the bridge’s modal frequencies can be measured by using only one accelerometer. Therefore, up to today, most FE model updating practices reported by literatures are only based on measured modal frequencies due to the limited field sensing resources. Therefore, to validate the effectiveness of the common practice, this article focuses on natural frequencies. However, some mode shapes have fortunately been measured on location and reported in Ref. [9]. Therefore, we have additionally utilized this information to further validate the accuracies of the models already updated using RS method and MSVR method based on the natural frequencies. A new section (Subsec. 4.3) has been added accordingly. The added work suggests that the MSVR method is superior to the RS method, which cannot be reflected from the natural frequencies (Subsec. 4.2) and the static structural responses (Sec. 5) calculated from the updated FE models.

  1. The details to establish the RS and MSVR models are missing. This is the critical step because of the trade-off between efficiency and accuracy. how to do the design of experiment (DOE) to generate samples for constructing the surrogate models in this research?

Response: Yes, according to Ref. [27], the utilized sampling method is more important than the surrogate model type used in the context of the surrogate model–based dynamic FE model updating. We choose CCD experimental design to generate samples for constructing the surrogate models, because Ref. [27] suggests that the quality of model updated using the CCD sampling method is higher than that obtained with other sampling techniques. This point has been emphasized in the revised manuscript (see the part highlighted in green).

  1. Authors mentioned “optimization” used in the model updating procedure. Did the authors just utilize the optimization toolbox in numerical software or develop the in-house optimization code? Those information needs to be added.

Response: Thank you for the comment. We use the optimization toolbox embedded in software Design-Expert (Version 6.0.10) for the RS method, instead of develop the in-house optimization code. This is explained in the revised manuscript (see the part highlighted yellow).

  1. I would suggest enlarging the label size in Figures 10 and 14. The fonts are too small to read. In Figure 10, can authors use scientific notations instead of the real value?

Response: Thank you. These figures have been revised accordingly.

  1. The title of the work includes the keyword “fatigue analysis”. However, in section 7, when the authors talked about the fatigue analysis, the theory used all is from the other’s work and no specific result is given. The work cited is a Ph.D. dissertation dated back to 2009, which is quite old. This needs to be carefully explained. Otherwise, the contribution of the work becomes questionable, and the title somehow should be changed since the fatigue analysis actually is not conducted.

Response: Thank you. As suggested, the title has been changed to “Structural Health Monitoring-Oriented Finite Element Model for a Special-Shaped Steel Arch Bridge and Its Application”. Now we steer away from the keyword “fatigue analysis” in the title, since the fatigue analysis actually is not the focus of the present article.

Reviewer 2 Report

Pls. see attached file.

Author Response

Reviewer 2

  1. Minor recommendations for the improvement of the manuscript

- Please reformulate the first phrase from the abstract section, as a bridge is not “suffering”. An alternative reformulation could be “…of a steel bridge subjected to vehicle-generated fatigue…”.

Response: Thank you. We have reformulated the phrase as suggested.

- Please maintain the same line spacing within the manuscript (Rows 344- 386 seem to have a higher line spacing, compared to the rest of the content).

Response: Thank you. We have revised the line spacing to make it consistent throughout the manuscript.

- In most of the Eq.’s are not explained all the terms. Please revise accordingly.

Response: Thanks for the comment. All terms in all the Eqs. have been explained at the first time they appear in the revised manuscript.

  1. Major recommendations for the improvement of the manuscript:

There are following drawbacks to the manuscript:

- At the end of Section 1, the main contributions of the paper should be better explained. There is lack of comparison with the literature. In particular, it is essential that the authors demonstrate the quality or efficiency of their results, compared to well-established methods. Furthermore, the Introduction section should be extended, because in the actual state it is to poor.

Response: Thanks. As suggested, the Introduction part has been extended with the novelty of the work and concrete contents discussing the state-of-the-art (see the parts highlighted in red and blue). Besides, we have discussed the well-established FE model updating method [1,29,30] and compared it with the new approaches utilized by the present study in the revised Introduction. The main contributions of the present study are thereby demonstrated.

- Please make sure that the picture shown in figure 1a is not subjected to copyright protection.

Response: Thank you for the suggestion. We have checked this figure and are sure that this figure is not subjected to copyright issue.

- The Conclusion section is sooner a summary of the work, without really presenting the conclusions that can be drawn from this research. Therefore, the quantitative results are required, and the meaningfulness of this study would be emphasized rather than presenting a summary on the technical works. Furthermore, the section should be re-written more comprehensively and it has to be extended with:

✓ the weak points of the proposed method;

✓ limitations of the method;

✓ further studies.

Response: Thank you for the good comment. Yes, the Conclusion section is fairly weak before this round of revision. To enhance this part, we have reported some important quantitative results in it (see the parts highlighted in orange), and extended it with the weak points of the proposed method, limitations of the method, and further studies (see the parts highlighted in pink).

- 16 of the 20 references (80%) are older the 10 years. Thus, the actual References are not able to support the idea of the actuality of the treated subject. Therefore, a major revision of the References is mandatory, in order to ensure that most of the titles are actual and not older than 10 years.

Response: Thank you for the comment. To reflect the state of the art of the knowledge related to present scientific issue, 8 significant articles published in 2021 and 2022 have been added in our reference list [21-28]. We suppose the present references can support the idea of the actuality of the treated subject. Thank you for your good suggestion.

Round 2

Reviewer 1 Report

The authors' revisions have addressed my comments.

Reviewer 2 Report

Manuscript may be accepted in present form